# Open-top Bessel beam two-photon light sheet microscopy for three-dimensional pathology

Won Yeong Park[1], Jieun Yun[1], Jinho Shin[2], Byung Ho Oh[3], Gilsuk Yoon[4], Seung-Mo Hong[5], Ki Hean Kim[1,6,7]*

[1]Department of Mechanical Engineering, Pohang University of Science and Technology, Pohang, Republic of Korea; [2]Department of Medicine, University of Ulsan College of Medicine, Seoul, Seoul, Republic of Korea; [3]Department of Dermatology, College of Medicine, Yonsei University, Seoul, Republic of Korea; [4]Department of Pathology, School of Medicine, Kyungpook National University, Daegu, Republic of Korea; [5]Department of Pathology, Asan Medical Center, University of Ulsan College of Medicine, Seoul, Republic of Korea; [6]Medical Science and Engineering Program, School of Convergence Science and Technology, Pohang University of Science and Technology, Pohang, Republic of Korea; [7]Institute for Convergence Research and Education in Advanced Technology, Yonsei University, Seoul, Republic of Korea

*For correspondence:
kiheankim@postech.ac.kr

Competing interest: The authors declare that no competing interests exist.

## eLife assessment

This **important** work by Park et al. demonstrates an open-top two-photon light sheet microscopy (OT-TP-LSM) for lesser invasive evaluation of intraoperative 3D pathology. The authors provide **convincing** evidence for the effectiveness of this technique investigating various human cancer cells. This article will be of broad interest to biologists and, specifically, pathologists utilizing 3D optical microscopy.

**Abstract** Nondestructive pathology based on three-dimensional (3D) optical microscopy holds promise as a complement to traditional destructive hematoxylin and eosin (H&E) stained slide-based pathology by providing cellular information in high throughput manner. However, conventional techniques provided superficial information only due to shallow imaging depths. Herein, we developed open-top two-photon light sheet microscopy (OT-TP-LSM) for intraoperative 3D pathology. An extended depth of field two-photon excitation light sheet was generated by scanning a nondiffractive Bessel beam, and selective planar imaging was conducted with cameras at 400 frames/s max during the lateral translation of tissue specimens. Intrinsic second harmonic generation was collected for additional extracellular matrix (ECM) visualization. OT-TP-LSM was tested in various human cancer specimens including skin, pancreas, and prostate. High imaging depths were achieved owing to long excitation wavelengths and long wavelength fluorophores. 3D visualization of both cells and ECM enhanced the ability of cancer detection. Furthermore, an unsupervised deep learning network was employed for the style transfer of OT-TP-LSM images to virtual H&E images. The virtual H&E images exhibited comparable histological characteristics to real ones. OT-TP-LSM may have the potential for histopathological examination in surgical and biopsy applications by rapidly providing 3D information.

## Introduction

Precise intraoperative cancer diagnosis is crucial for achieving optimal patient outcomes by enabling complete tumor removal. The standard method is the microscopic cellular examination of surgically excised specimens following various processing steps, including thin sectioning and hematoxylin and eosin (H&E) cell staining. However, this examination method is laborious and time-consuming. Furthermore, it has inherent artifacts that disturb accurate diagnosis, including tissue loss, limited two-dimensional (2D) information, and sampling error (*Liu et al., 2021*). High-speed three-dimensional (3D) optical microscopy, which can visualize cellular structures without thin sectioning, holds promise for nondestructive 3D pathological examination as a complement of 2D pathology limitation (*Liu et al., 2021*; *Almagro et al., 2021*; *Bishop et al., 2022*; *Serafin et al., 2023*). Various high-speed 3D microscopy techniques have been developed (*Filler and Peuker, 2000*; *van Royen et al., 2016*; *Verhoef et al., 2019*; *Yoshitake et al., 2016*; *Tao et al., 2014*; *Baugh et al., 2017*; *Patel et al., 2022*; *Voleti et al., 2019*), and open-top light sheet microscopy (OT-LSM) is a promising technique that enables high-throughput imaging with light sheet illumination and planar imaging on uneven tissue surfaces (*McGorty et al., 2015*; *Glaser et al., 2017*; *Glaser et al., 2022*; *Gao et al., 2023*). OT-LSM is a specialized form of LSM for large tissue specimens such as biopsy specimens with all the optical components placed below the sample holder. However, fluorescence 3D microscopy techniques including OT-LSM have shallow imaging depths in turbid tissue owing to light scattering and absorption, limiting the applicability of 3D histopathology. As a deep tissue imaging method, two-photon microscopy (TPM) has been used in both biological and optical biopsy studies (*Denk et al., 1990*; *So et al., 2000*; *Helmchen and Denk, 2005*). TPM is based on nonlinear two-photon excitation of fluorophores and achieves high imaging depths down to a few hundred micrometers by using long excitation wavelengths, which reduce light scattering. Moreover, TPM provides additional intrinsic second harmonic generation (SHG) contrast for visualizing collagen fibers within the extracellular matrix (ECM). This feature proved advantageous for high-contrast imaging of cancer tissue and microenvironmental analysis (*Zipfel et al., 2003*; *Fast et al., 2020*; *Park et al., 2022*). However, TPM has low imaging speeds due to point scanning-based imaging. To address this limitation, two-photon LSM (TP-LSM) techniques were developed for high-speed imaging (*Fahrbach et al., 2013*; *Mahou et al., 2014*; *Wolf et al., 2015*; *Maioli et al., 2020*; *de Vito et al., 2022*). Although TP-LSM facilitated rapid 3D imaging of cancer cells and zebrafish, its applications were limited to small samples and biological studies due to geometric limitations. Conventional TP-LSM had a configuration of a horizontally oriented illumination objective and a vertically oriented imaging objective. This geometry imposed limitations on the sample size, rendering it unsuitable for the examination of centimeter-scale specimens. TP-LSM with open-top configuration is needed for 3D histological examination.

In this study, we present the development of an open-top TP-LSM (OT-TP-LSM) with a Bessel beam for high-throughput and high-depth 3D pathological examination of large tissue specimens. A nondiffractive Bessel beam was used and scanned in one-dimensional (1D) to generate an extended depth of field (EDOF) excitation light sheet with minimal side lobe effects (*Planchon et al., 2011*; *Fan et al., 2020*). High imaging depths were achieved by using long excitation wavelengths and fluorescent nuclear probes emitting relatively long wavelengths. After development and characterization, the system was applied to high-throughput 3D imaging of various human cancer specimens, including skin, pancreas, and prostate in comparison with conventional H&E stained histological images for verification. Moreover, OT-TP-LSM images were converted to virtual H&E images using the deep learning-powered style transfer. We demonstrated that OT-TP-LSM images were comparable to histological images of H&E stained slides from several human cancer types, and OT-TP-LSM 3D imaging might have the potential for rapid and accurate nondestructive 3D pathology.

## Results

### Open-top two-photon light sheet microscopy (OT-TP-LSM) for 3D pathology

Schematics and characterization results of OT-TP-LSM are presented in *Figure 1*. OT-TP-LSM is an orthogonally arranged dual objective LSM with a liquid prism interface (*Figure 1A*). All optical components of OT-TP-LSM were positioned underneath a sample holder to accommodate large size specimens without physical interference with the illumination and imaging optics. Two 20x air objective

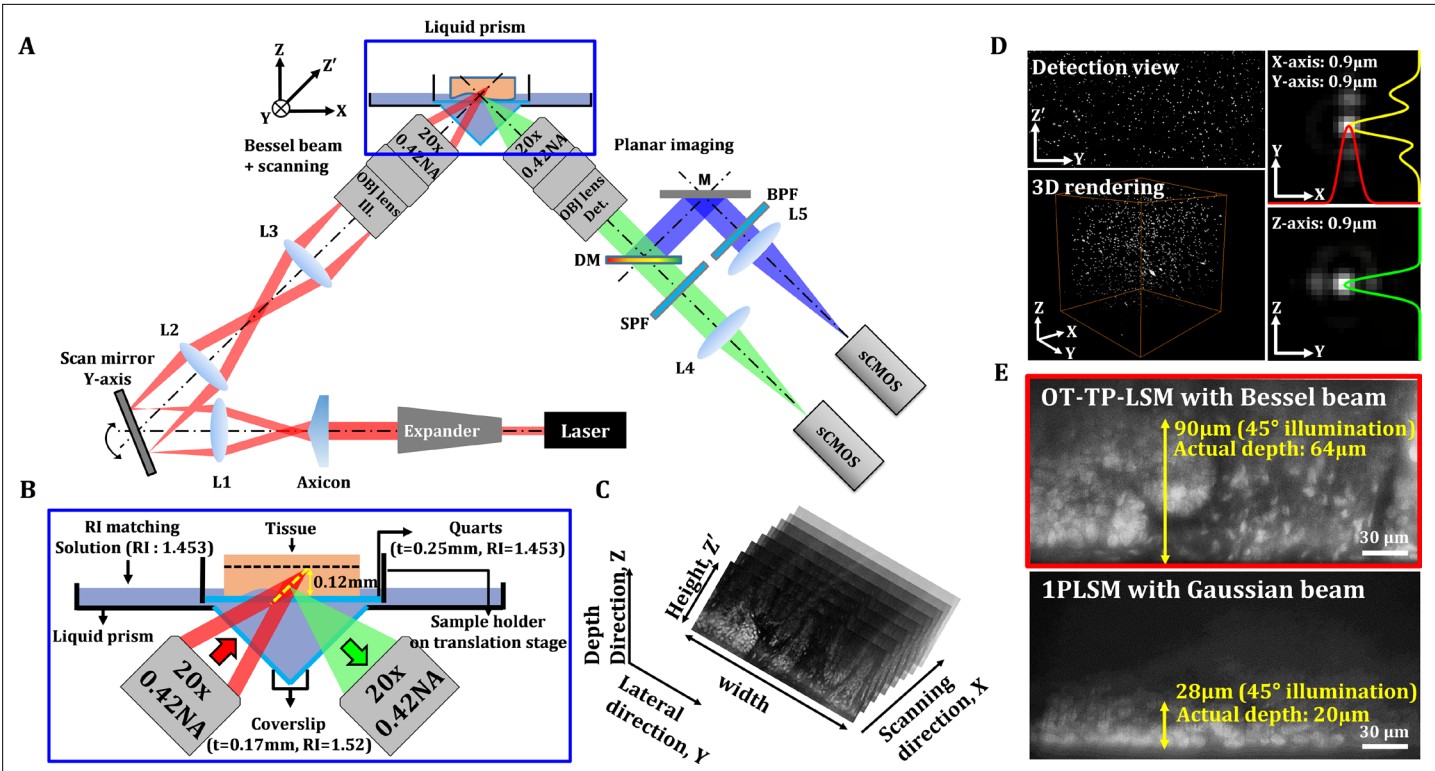

**Figure 1.** Schematics and characteristics of open-top two-photon light sheet microscopy (OT-TP-LSM). (**A**) Overall design of OP-TP-LSM. L: lens; M: mirror; DM: dichroic mirror; SPF: short pass filter; BPF: band pass filter; OBJ: objective lens. (**B**) Detailed schematic of the sample interface, including the liquid prism filled with refractive index (RI)-matching solution. (**C**) Illustration of sequential light sheet imaging with stepwise lateral translation. (**D**) Image resolution characterized by three-dimensional (3D) imaging of 0.5 μm fluorescent microspheres. (**E**) Imaging depth characterized by imaging a proflavine-labeled fresh human skin specimen in comparison with that of one-photon light sheet microscopy (1PLSM) using Gaussian excitation sheet.

The online version of this article includes the following figure supplement(s) for figure 1:

**Figure supplement 1.** Detail optical configuration of illumination arm in open-top two-photon light sheet microscopy (OT-TP-LSM).

lenses with a moderate numerical aperture (NA) of 0.42 were used to deliver a thin excitation light sheet and to collect emission light efficiently. The custom liquid prism was used to transmit excitation light into the specimen and to collect emission light out with minimal aberration by refractive index (RI) matching with different mediums where the sample was immersed (**Figure 1B**). The liquid prism was filled with diluted glycerol in distilled water, whose RI was matched to that of a quartz (RI: 1.45) window in a sample holder. A Ti-Sapphire laser beam was expanded and converted to a Bessel beam using an axicon lens. The Bessel beam was reflected on a galvanometer scanner and relayed to the sample through the liquid prism by a combination of a tube lens and the illumination objective lens. The scanner scanned the Bessel beam to form an EDOF excitation light sheet. Emission light generated by the sample was collected by the imaging objective lens through the liquid prism, split into two channels, one for fluorescence and the other for SHG, using a dichroic mirror, and imaged simultaneously by two scientific complementary metal-oxide–semiconductor (sCMOS) cameras with individual gain control.

The excitation light sheet was designed to have a DOF of 180 μm with a thickness of 0.9 μm (**Figure 1—figure supplement 1**). The imaging field of view was 600 μm × 170 μm with 1,850 × 512 pixels and was limited by the DOF of the Bessel beam. Emission light scattering in the specimen further limited the imaging depth. Selective planar images along the oblique excitation light sheet were acquired sequentially with a stepwise X-axis translation of the specimen by the motorized stage (**Figure 1C**). A graphical user interface based on LabVIEW was used to generate trigger signals for both the cameras and the motorized stage. OT-TP-LSM was characterized by imaging fluorescent microspheres (0.5 μm in diameter) embedded in agarose gel, whose RI was matched to that of the solution in the liquid prism (**Figure 1D**). The image throughput was 0.24 mm²/s at an acquisition rate

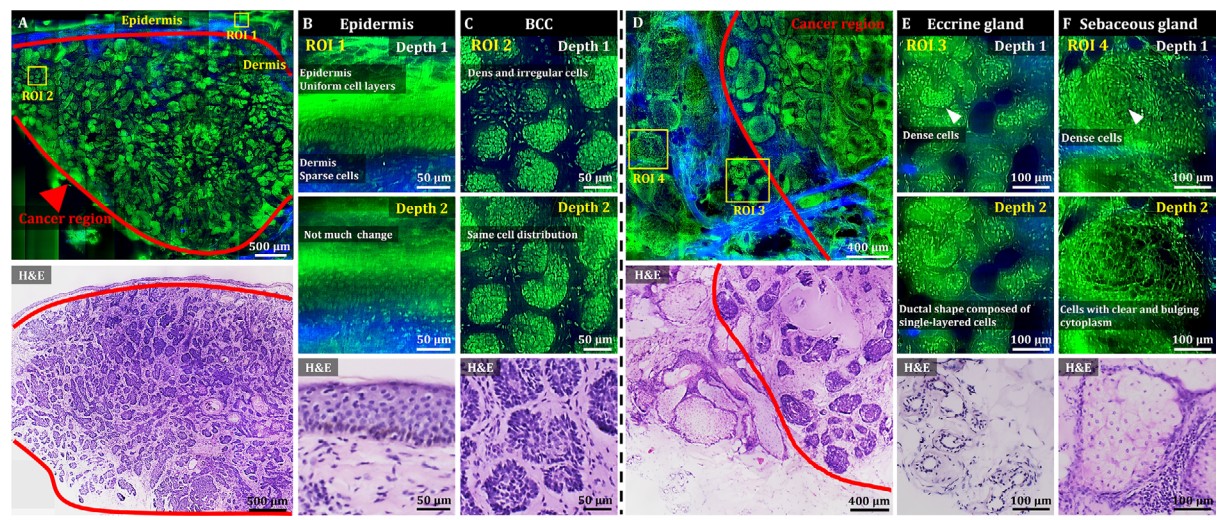

**Figure 2.** 3D OT-TP-LSM images of fresh human skin cancer (basal cell carcinoma, BCC) specimens. (**A**) Large-sectional OT-TP-LSM and hematoxylin and eosin (H&E) stained slide images of a BCC case 1. Red solid line indicates BCC area. (**B and C**) Two magnified images of regions of interest (ROIs) with corresponding H&E images. ROI 1 and 2 were in the epidermis and superficial dermis and the BCC region below, respectively. (**D**) Large-sectional OT-TP-LSM and H&E images of a BCC case 2. Red solid line indicates BCC area. (**E and F**) Two magnified images of ROIs with corresponding H&E images. ROI 3 and 4 were in eccrine and sebaceous glands, respectively. Superficial cell layers of both glands are marked with a white arrow. In the OT-TP-LSM images, proflavine fluorescence and second harmonic generation (SHG) are displayed in green and blue, respectively.

The online version of this article includes the following video for figure 2:

**Figure 2—video 1.** Three-dimensional (3D) visualization of basal cell carcinoma (BCC) structures and normal cell structures.
https://elifesciences.org/articles/92614/figures#fig2video1

---

of 400 fps and step size of 1 µm. The image resolution was measured to be 0.9 µm in all dimensions. The lateral resolution in the Y-Z' plane was approximately 128% of the theoretical value due to spherical aberration induced by the RI mismatch between air and the immersion medium, but this was sufficient for detecting cellular features.

The imaging depth of OT-TP-LSM was evaluated using freshly excised human skin specimens. Proflavine was topically instilled onto the specimen for nuclear labeling before imaging. A representative Y-Z' plane image is shown in *Figure 1E*. The imaging depth in the skin was approximately 90 µm in the 45°-tilted image plane, and the actual depth was 64 µm from the surface. To compare the imaging depth, the same skin specimen was imaged using conventional one-photon excitation-based LSM (1PLSM). The imaging depth was approximately 28 µm in the 45°-tilted image plane and 20 µm from the surface, which was approximately three times less than that of OT-TP-LSM. While the imaging depth of OT-TP-LSM was sufficient for visualizing 3D cell structures in the skin, it was relatively low compared to that of conventional point-scanning based TPM due to emission photon scattering.

## 3D OT-TP-LSM imaging of human skin cancers

After development and characterization, the performance of OT-TP-LSM was assessed in human skin cancer specimens. Representative OT-TP-LSM images of two basal cell carcinoma (BCC) specimens are presented along with H&E stained histology images in *Figure 2*. The H&E images were obtained from the same specimens at slightly different depths. The OT-TP-LSM images visualized both cell structures and ECM in the cross-sectioned skin specimens, where proflavine-labeled cell nuclei and SHG-emitting collagen were depicted in green and blue, respectively. A mosaic image of the first BCC specimen visualized the epidermis as a thin layer on the top and the underlying dermis, most of which was occupied by BCC nests (*Figure 2A*). The epidermis and dermis were easily identifiable with SHG contrasts. Two regions of interest (ROI) were selected in the epidermis and BCC, and magnified images of these ROIs are presented to show the detailed cell structures. Magnified image in the epidermis (ROI 1) showed layered keratinocytes (*Figure 2B*). Keratinocytes in the basal layer were relatively large and individually resolved, while those in the upper layers were unresolved and

appeared as a band. It could be attributed to the upper layers being comprised of flat cells with relatively small cytoplasm, resulting in little space between nuclei. Additionally, strong autofluorescence signal in the stratum corneum might prevent visualization of the cells in the superficial layer. Magnified images of BCC (ROI 2) showed many irregularly shaped tumor nests composed of densely packed monomorphic tumor cells of the same size (*Figure 2C*).

The mosaic image of the second BCC specimen mainly showed two different cell structures in the dermis (*Figure 2D*). The structures on the right side were BCCs, while those on the left side had intact ECM composition and differed from BCCs. The corresponding H&E stained histology image confirmed these as BCCs and normal gland structures in the dermis, respectively. Two ROIs were selected in the nontumorous skin region, and magnified images at two different depths are presented along with their corresponding H&E stained images. Magnified images of ROI 3 visualized clusters of small round tubular structures (*Figure 2E*). The image at depth 2 showed a ductal structure composed of single-layered cuboidal epithelial cells, which is typical of eccrine glands. Magnified images of ROI 4 at two different depths showed pear-shaped clusters consisting of cells with clear and bulging cytoplasm, which is typical of sebaceous glands (*Figure 2F*). Using 3D visualization, normal glandular structures in the dermis were distinguished from BCC tumor nests (*Figure 2—video 1*). Both eccrine and sebaceous glands could appear similar to BCC nests in 2D images at certain depths. Hence, nondestructive 3D visualization of cell structures would be important for distinguishing them, serving as a complement to the traditional 2D H&E images.

## 3D OT-TP-LSM imaging of human pancreatic cancers

We then used OT-TP-LSM to visualize human pancreas specimens including those with pancreatic cancer. Instead of proflavine, propidium iodide (PI) was used as a nuclear labeling agent to increase the imaging depth (*Figure 3—figure supplement 1*). The proflavine-labeled pancreas specimens

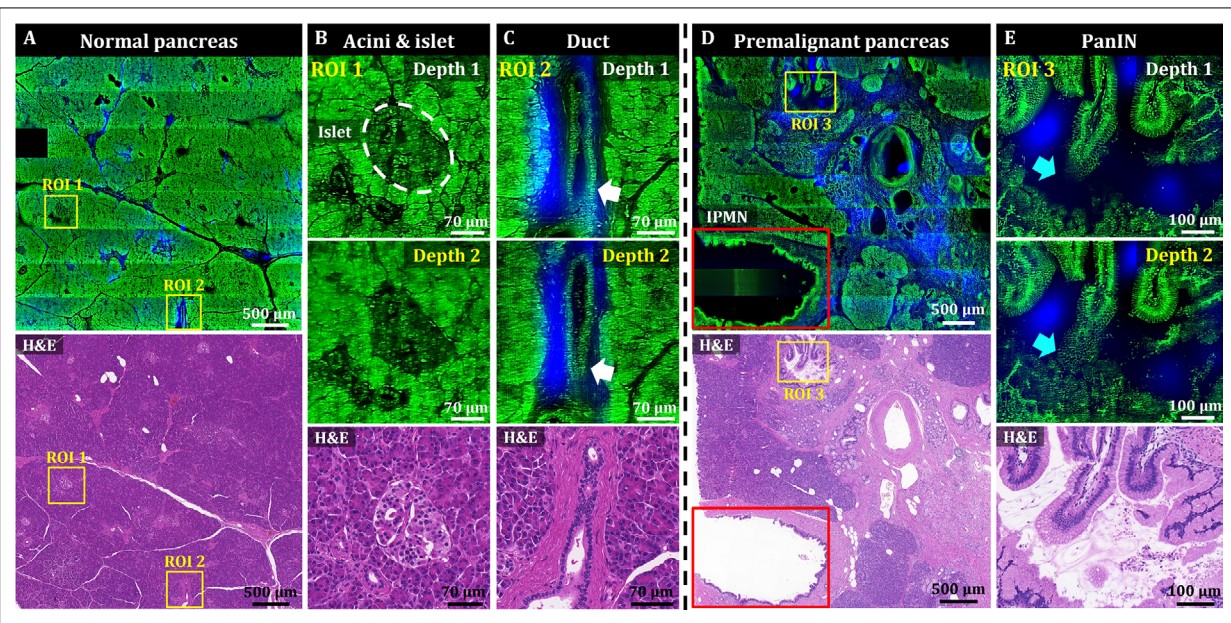

**Figure 3.** 3D OT-TP-LSM images of two human pancreas specimens: normal and premalignant lesion. (**A**) Large-area OT-TP-LSM and H&E stained images of normal human pancreas. (**B and C**) Magnified two-depth OT-TP-LSM images of ROI 1 and 2 in the normal pancreas and the corresponding H&E stained slide images. ROI 1 and 2 were on the acini and islets of Langerhans and the pancreatic duct, respectively. 3D morphological change of the duct was marked with a white arrow. (**D**) Large-area OT-TP-LSM and H&E stained images of a pancreatic premalignant lesion. Red box indicates the intraductal papillary mucinous neoplasm (IPMN). (**E**) Magnified two-depth OT-TP-LSM images of ROI 3 and corresponding H&E image of pancreatic intraepithelial neoplasia (PanIN). 3D morphological change of papillary in the PanIN was marked with a cyan arrow. In OT-TP-LSM images, propidium iodide (PI)-labeled cells and SHG-emitting collagen are displayed in green and blue, respectively.

The online version of this article includes the following figure supplement(s) for figure 3:

**Figure supplement 1.** Comparison of imaging depth between propidium iodide (PI) and proflavine in human pancreas.

**Figure supplement 2.** PI cell staining characteristics in human pancreas.

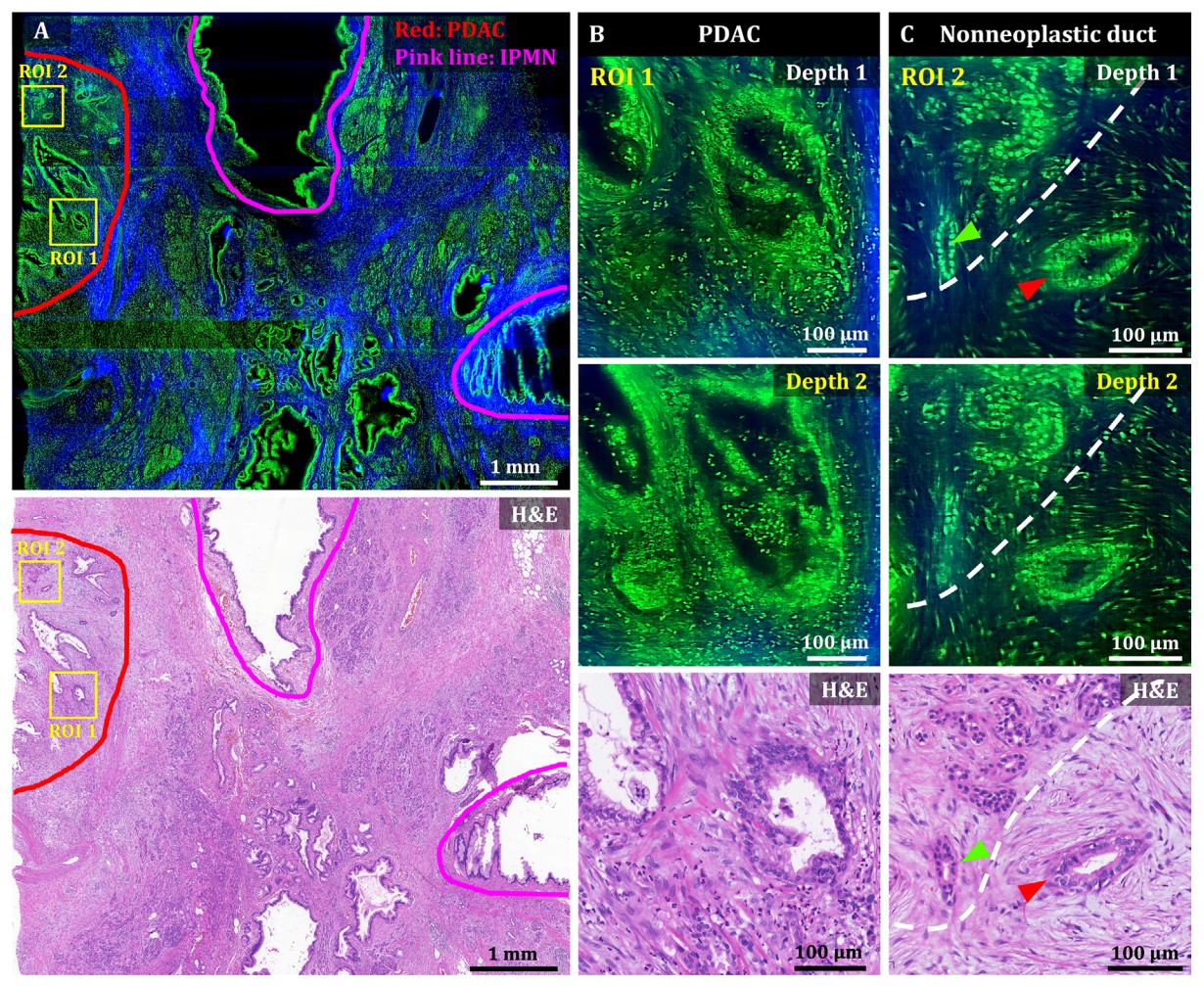

**Figure 4.** 3D OT-TP-LSM images of human pancreatic ductal adenocarcinoma (PDAC). (**A**) Large-area OT-TP-LSM and H&E stained slide images of PDAC arising from IPMN specimen. Red and pink solid line indicates the PDAC and the IPMN area, respectively. (**B and C**) Magnified two-depth OT-TP-LSM images of ROI 1 and 2 in the PDAC region and the corresponding H&E stained images. ROI 1 was on the PDAC, and ROI 2 was on the boundary region between nonneoplastic pancreatitis and PDAC, respectively. A nonneoplastic duct and a PDAC were marked with green and red arrows, respectively. A white dashed line indicated the boundary between nonneoplastic pancreatitis and PDAC. In the OT-TP-LSM images, PI-labeled cells and SHG-emitting collagen are displayed in green and blue, respectively.

had a relatively shallow imaging depth (up to 30 μm approximately) owing to the dense desmoplastic stromal cell composition. PI labeling allowed for deeper imaging up to approximately 55 μm from the surface due to the longer peak emission wavelength of 640 nm compared to proflavine. PI cytoplasmic labeling also occurred in addition to nuclear labeling of both acinar and neoplastic cells, probably owing to PI labeling of both DNA and RNA (*Figure 3—figure supplement 2* ; *Rieger et al., 2010*; *Niu et al., 2015*). Cell nuclei were identified with relatively strong fluorescence in the membrane and weaker fluorescence in the core. Representative OT-TP-LSM images and corresponding H&E stained slide images of normal pancreas and premalignant lesion specimens, and a pancreatic cancer (also known as pancreatic ductal adenocarcinoma, PDAC) specimen are shown in *Figures 3 and 4*, respectively. The mosaic image of the normal pancreas visualized a dense distribution of acini composed of exocrine acinar cells in good correlation with the corresponding H&E image (*Figure 3A*). Two ROIs were selected, and magnified images are presented to show detailed cytologic features of the normal pancreas. Magnified images of ROI 1 visualized the cellular structure of acini and islets of Langerhans expressing different fluorescence intensities (*Figure 3B*). Acini were composed of lobules of acinar cells, and the boundaries of the acini were identified with relatively low PI fluorescence. The islets of Langerhans expressed relatively low PI fluorescence, probably due to the high composition of

crystallized insulin in beta cells (*Lemaire et al., 2012*). Magnified images of ROI 2 visualized a duct in the normal pancreas (*Figure 3C*). The duct was composed of single-layered cuboidal epithelial cells with round nuclei and was surrounded by dense fibrous tissue expressing SHG. The mosaic image of the pancreatic premalignant lesion exhibited different and irregular structures, including dilated ducts with intraductal proliferation of papillary structures (*Figure 3D*). Two different types of precursor lesions, intraductal papillary mucinous neoplasm (IPMN) and pancreatic intraepithelial neoplasia (PanIN), were observed (*Hruban et al., 2001*; *McGinnis et al., 2020*), and tumor cells were tall columnar containing intracytoplasmic mucin (*Adsay et al., 2004*). A corresponding H&E histology image confirmed the precursor lesions. Magnified images of PanIN (ROI 3) at two different depths showed proliferation of papillary epithelial cells with intracytoplasmic mucin, and 3D morphology of the PanIN (*Figure 3E*). OT-TP-LSM images were very similar to the corresponding H&E images, indicating OT-TP-LSM provided comparable histopathological information to that of H&E stained slides without thin sectioning.

A mosaic image of the PDAC arising from the IPMN specimen showed various abnormal ductal structures including cancer glands, IPMN, and surrounding pancreatitis, which were confirmed by the corresponding H&E stained slide image (*Figure 4A*). More desmoplastic fibrous stroma and a few abnormal cancer glands were observed compared to normal pancreas tissue (*Drifka et al., 2015*; *Ray et al., 2022*). Two ROIs were selected in the PDAC region and the boundary region with nonneoplastic pancreatitis, and magnified images are presented to show detailed cellular structures. Magnified images of ROI 1 (PDAC) at two different depths showed irregularly shaped glands with sharp angles and 3D structural complexity including unstable bridging structure inside (*Figure 4*). An irregular and distorted architecture amidst desmoplastic stroma is one of the important diagnostic factors for PDAC *Adsay et al., 2004*. The cancer glands exhibited disorganized cancer cell arrangement with nuclear membrane distortion. Magnified images of ROI 2 showed both nonneoplastic ducts and cancer glands in different cell arrangements (*Figure 4C*). The nonneoplastic ducts showed single-layered epithelium with small, evenly distributed cells expressing relatively high nuclear fluorescence. Cancer glands, on the other hand, had disorganized and multilayered structure with large nuclei. OT-TP-LSM visualized

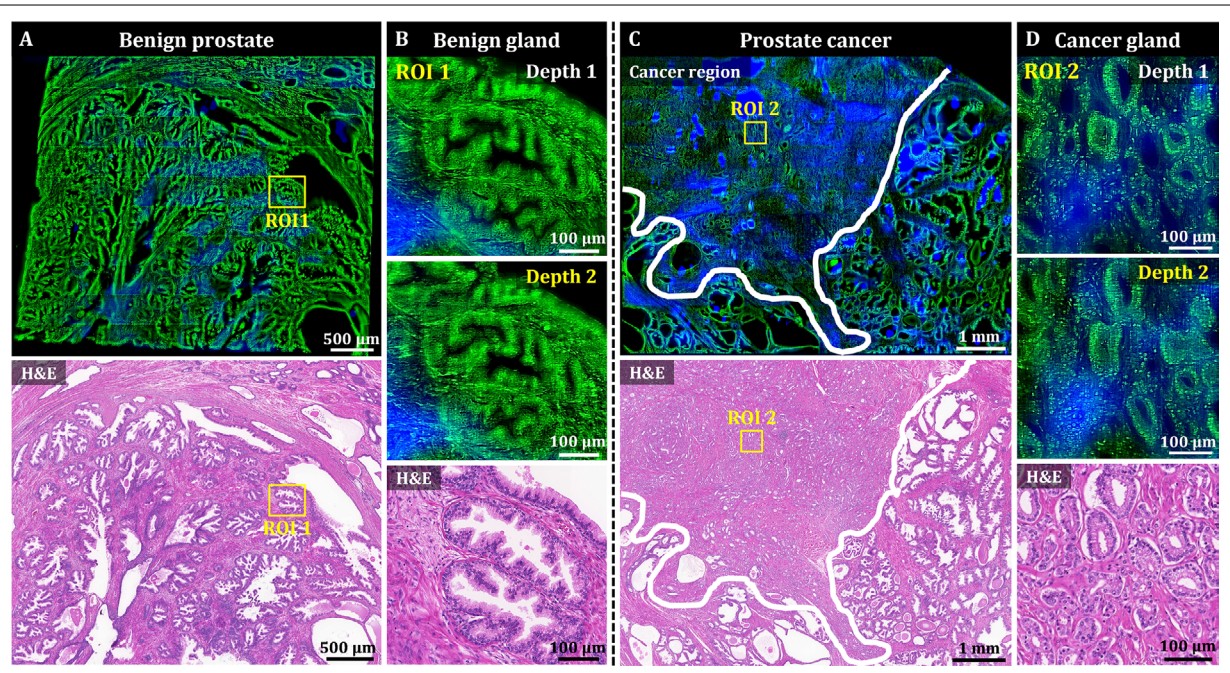

**Figure 5.** 3D OT-TP-LSM images of human prostate specimens: benign and adenocarcinoma. (**A**) Large-area OT-TP-LSM and H&E stained slide images of a benign prostate specimen. (**B**) Magnified two-depth OT-TP-LSM images of ROI 1 in the benign gland and a corresponding H&E image. (**C**) Large-area OT-TP-LSM and H&E images of a prostatic cancer specimen. A white solid line indicated the boundary between adenocarcinoma and benign regions. (**D**) Magnified two-depth OT-TP-LSM images of ROI 2 in adenocarcinoma glands and a corresponding H&E image. In OT-TP-LSM images, PI-labeled cells and SHG-emitting collagen were visualized in green and blue, respectively.

the 3D invasiveness of cancer glands within tissues nondestructively, which could not be identified from limited 2D information.

## 3D OT-TP-LSM imaging of human prostatic cancers

OT-TP-LSM was used to visualize human prostatectomy specimens, and two representative OT-TP-LSM images of benign and adenocarcinoma specimens are presented in *Figure 5*. The mosaic image of the histologically confirmed benign prostate showed benign glands dispersed throughout the specimen with relatively weak SHG composition between the glands (*Figure 5A*). An ROI was selected on benign glands, and magnified images are presented to show cytologic features alongside the corresponding H&E stained slide image (*Figure 5B*). Multiple cell layers, consisting of secretory cells and basal cells, lined the central lumen in the ducts of benign glands. Although images at two different depths are presented, the cell distribution was similar because of the relatively small depth difference compared to the gland size. The mosaic image of the prostate adenocarcinoma in *Figure 5C* showed both the benign and cancer regions, whose boundary was marked with a white solid line, and they were confirmed by the corresponding H&E stained image. The cancer region was full of crowded small gland structures, whereas the benign region consisted of relatively large glands surrounded by fibrous stroma. Magnified images of ROI 2 in the cancer region showed small glands composed of monolayered cell walls (*Figure 5D*). The cancer glands had single-layered malignant secretory cells only due to the loss of basal cell layers. The corresponding H&E histology was diagnosed as Gleason score 3+3 prostatic adenocarcinoma. OT-TP-LSM provided histological 3D information equivalent to that of the H&E stained image without the need for sectioning.

## Deep learning-based style transfer for virtual H&E

OT-TP-LSM images visualized histological features with high similarity to those in H&E images, but they may be unfamiliar to pathologists and clinicians accustomed to traditional H&E stained slide histology. PI-labeled OT-TP-LSM images were converted to virtual H&E images using Cycle-consistent generative adversarial network (CycleGAN; *Zhu et al., 2017*). Representative virtual H&E images of pancreatic premalignant (PanIN) and PDAC are presented along with the corresponding real H&E stained images for comparison in *Figure 6*. Mosaic virtual H&E images of both PanIN and PDAC showed similar patterns to the corresponding real H&E stained histology images (*Figure 6A and C*). Although the normal region in the lower left corner of the virtual PanIN image was not transformed properly possibly due to insufficient image data of normal acini for training (*Figure 6A*), the magnified virtual H&E images clearly displayed distinct cell distributions of PanIN and PDAC by labeling

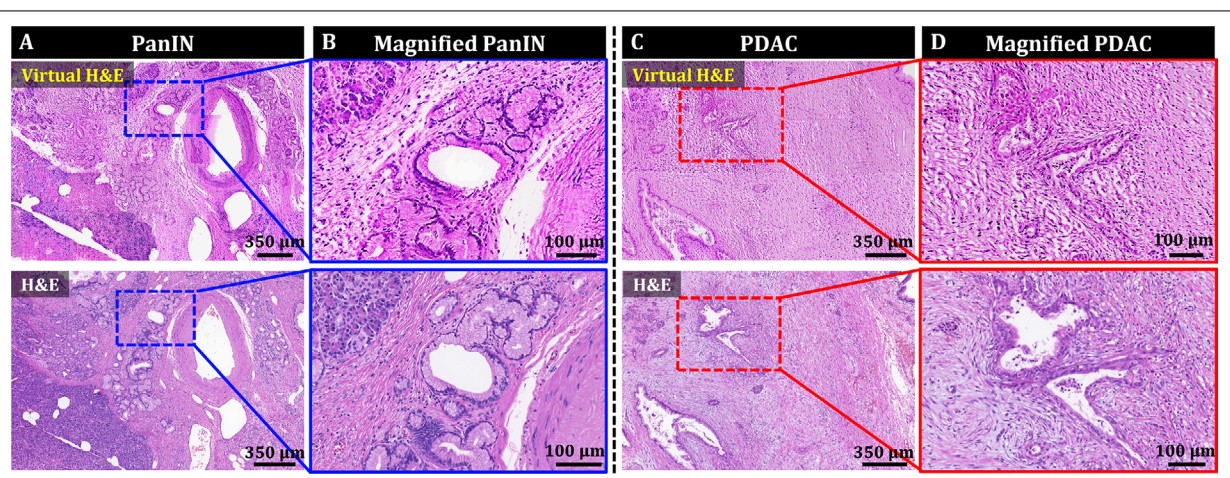

**Figure 6.** Style transfer of OT-TP-LSM images for virtual H&E by cycle-consistent generational adversarial networks (CycleGAN). (**A and C**) Large-area virtual H&E images and corresponding H&E stained slide images of PanIN and PDAC specimens, respectively. (**B and D**) Magnified virtual H&E images and corresponding H&E stained images of PanIN and PDAC, respectively.

The online version of this article includes the following figure supplement(s) for figure 6:

**Figure supplement 1.** Cycle-consistent generational adversarial networks (CycleGAN) workflow for style transfer of OT-TP-LSM image to virtual hematoxylin and eosin (H&E) image.

cell cytoplasm and nuclei in pink and purple, respectively (*Figure 6B and D*). PanINs were visualized as slightly dilated, irregularly clustered ducts with preserved structural integrity (*Figure 6B*), whereas PDAC had disorganized cell arrangement with haphazard patterns, infiltrating into the surrounding tissue with desmoplastic stroma (*Figure 6D*). Histological features found in the virtual H&E images were diagnostic factors for the discrimination of PDAC and PanIN. Virtual H&E staining of OT-TP-LSM via CycleGAN could provide comparable cellular information to conventional H&E stained histology.

## Discussion

OT-TP-LSM was developed for the rapid and precise nondestructive 3D pathological examination of excised tissue specimens during both biopsy and surgery, as a compliment to traditional 2D H&E pathology by visualizing 3D cell structures. OT-TP-LSM generated a two-photon excitation light sheet using one-dimensional Bessel beam scanning, and planar images were acquired by collecting emission light from the excitation light sheet with sCMOS cameras at 400 fps maximum. Proflavine and PI were used for nuclear labeling and cell structure visualization, and intrinsic SHG contrast was used for ECM visualization. The performance of OT-TP-LSM was characterized in comparison with histopathologic images of H&E stained slides in various human cancer specimens including skin, pancreas, and prostate. 3D OT-TP-LSM images were well matched with the corresponding 2D H&E histological images in all the human tissue specimens evaluated. The relatively high imaging depths of OT-TP-LSM enabled the nondestructive visualization of detailed 3D cell structures with high contrast and without distortion and allowed a distinction between cancer and normal cell structures as well as the detection of cancer invasiveness within tissues. These have been challenging with 2D histological images. The high-quality images of OT-TP-LSM might be partially attributed to the use of the nondiffractive Bessel beam for the generation of an EDOF excitation light sheet (*Olarte et al., 2012*).

OT-TP-LSM could visualize 3D cell structures in the skin down to 60 μm deep from the surface by using long wavelength excitation light and proflavine labeling. Although the imaging depth was not high compared to the one of conventional TPM, it was sufficient to visualize 3D cell structures in the skin without sectioning artifacts. The imaging depth varied depending on the tissue type, and the imaging depths of both pancreas and prostate were relatively low in comparison to that of the skin. This low imaging depth could be attributed to the dense cell composition and higher RI than the current immersion medium. To increase the imaging depth, PI with a relatively long emission wavelength was used. Additionally, an immersion medium with 1.5 RI could improve the imaging depth (*Young et al., 2011*), and a quick optical clearing method could also be used to enhance the imaging depth (*Aoyagi et al., 2015*).

We trained and applied CycleGAN to PI-labeled OT-TP-LSM images of pancreas specimens to generate virtual H&E images, and the cytoplasm and cell nuclei, which exhibited intensity differences in greyscale, were transformed to H&E-like pseudo colors. Virtual H&E images in pancreatic cancer provided histological features equivalent to the conventional H&E. Although CycleGAN showed the potential for virtual H&E staining, further development is needed to have robust performance and universal application. Therefore, CycleGAN will be trained by using more OT-TP-LSM images from various tissue types such as skin and prostate.

The current OT-TP-LSM has several limitations. One issue is spherical aberration arising from the RI mismatch between the liquid prism and air by using air objective lenses in both illumination and imaging arms. Spherical aberration degraded the image resolution, particularly in the imaging arm. An immersion objective lens with a high NA would be desirable to improve both image resolution and emission light collection efficiency. The excitation light sheet was produced by swiftly scanning a single Bessel beam, yielding a shorter pixel dwell time compared to that of a traditional single-photon 2D light sheet. Employing multiple Bessel beams for excitation light sheet generation would enhance pixel dwell time and image contrast [*Minamikawa et al., 2009*; *Chen et al., 2016*]. The relatively short excitation wavelength of 770 nm was used for all the imaging to efficiently collect intrinsic SHG light [20], but the imaging depth was limited. Longer excitation wavelengths with long emission wavelength probes would increase the imaging depth beyond those presented here. Although OT-TP-LSM enabled high-speed 3D imaging, the post-processing time of the OT-TP-LSM image datasets was relatively long due to the large data size, sequential processing of dual channel images, and manual stitching. The long post-processing time needs to be resolved for intraoperative applications. To speed up processing, these processing steps can be performed using field-programmable gate array

(FPGA)-based data acquisition with graphics processing unit (GPU)-based computing. The processing time can be further reduced by coding the algorithm in a C++-based environment. Furthermore, ImageJ-based software such as the Bigstitcher plugin can be used for automatic 3D image processing (*Hörl et al., 2019*).

In conclusion, OT-TP-LSM was developed for high-throughput and high-depth tissue examination. OT-TP-LSM was tested with various human cancers including skin, pancreas, and prostate by visualizing 3D cell structures. Intrinsic SHG contrast was used for visualizing ECM and detecting fibrosis. Normal glands and BCC nests were clearly distinguished, and both pancreatic and prostate cancers were clearly identified using 3D visualization of cell structures with OT-TP-LSM. Furthermore, virtual H&E staining of OT-TP-LSM using CycleGAN provided histopathologic information comparable to that from H&E stained slides. Therefore, OT-TP-LSM could be useful for rapid and precise nondestructive 3D pathology.

## Materials and methods

### OT-TP-LSM setup

The schematics of OT-TP-LSM are shown in *Figure 1A*. As the light source, OT-TP-LSM used a titanium–sapphire pulse laser (Chameleon II, Coherent) with 140-fs pulse width and 80-MHz pulse repetition rate. The excitation wavelength was set at 770 nm for imaging human tissue specimens. In the illumination arm, the laser beam was expanded from 1.4 mm to 1.86 mm and then converted to a Bessel beam by using an axicon lens (131-1270 + ARB825, Eksma). The Bessel beam was relayed onto the specimen through two lens pairs of L1 (AC254-045-B, Thorlabs) and L2 (AC254-080-B, Thorlabs), and L3 (AC254-150-B, Thorlabs) and a 20× air illumination objective lens with 0.42 NA (MY20X-804, Mitutoyo). The Bessel beam diameter at the back aperture of the objective lens was approximately 6.8 mm to prevent beam clipping, and the effective illumination NA was 0.34. The excitation light sheet was generated by Y-axis scanning of the Bessel beam with a galvanometric scanner (GVS102, Thorlabs), which was positioned between L1 and L2. The liquid prism transmitted the excitation light normally through a coverslip (HSU-0101122, Marienfeld Superior), RI-matching solution, and a 0.25 mm thick round quartz window (26043, Ted Pella) at the bottom of the sample holder, and then to the sample (Fig. 1B). Diluted 85% glycerol (Samchun) in distilled water was used for RI-matching with the quartz window (RI: 1.45). The sample holder was connected to an x-y motorized translation stage (XY MS-2000, Applied Scientific Instrumentation) for sample translation during imaging. To remove air gaps between the quartz window of the sample holder and irregular-shaped tissue specimen, the specimen was partially immersed in the RI-matching solution and light pressure was applied on the top of the specimen. In the imaging arm, emission light was collected by the imaging objective lens (MY20X-804, Mitutoyo) through the liquid prism, spectrally separated into two channels by a dichroic mirror (T425lpxr, Chroma), and collected at two sCMOS cameras (pco-edge 4.2, PCO) after passing through a tube lens (AC254-200-A, Thorlabs), respectively. SHG light was collected in the short pass channel with an emission filter (ET376/30x, Chroma), and fluorescence was collected in the long pass channel with an emission filter (ET680SP, Chroma). Both fluorescence and SHG images were simultaneously acquired with individual gain control.

### Human skin cancer specimen collection and imaging protocol

Fresh skin cancer specimens were collected under the Institutional Review Board (IRB) of Yonsei University, Severance Hospital, approved protocol (approval # 4-2021-0128). The study was conducted according to the Declaration of Helsinki Principles. OT-TP-LSM imaging was conducted within 12 hr post-excision. Patient information was deidentified. Before imaging, fresh skin specimens were cut cross-sectionally to approximately 4 mm thickness and topically instilled with 400 µg/mL proflavine solution (Sigma) for 5 min, and then rinsed in phosphate-buffered saline. OT-TP-LSM imaging was conducted in 3D on the sectioned surface. After imaging, the skin specimens were fixed and processed for histopathological evaluation with H&E stained slides. Microscopic images were obtained by using a wide-field microscope (Leica Z16 APO, Leica Microsystems).

## Human pancreatic and prostate cancer specimen collection and imaging protocol

Formalin-fixed, paraffin-embedded (FFPE) tissue blocks of pancreatic and prostate cancers were collected under approval (approval #, 2021-0393) from the IRB of Asan Medical Center, University of Ulsan College of Medicine. Four μm thickness sections were made for H&E stained slides, and the remaining blocks were used for OT-TP-LSM imaging. The FFPE tissue blocks were deparaffinized with xylene (99%, Daejung) solution overnight, 1:1 xylene-100% methanol (Optima grade, Fisher scientific) mixture solution for 1 hr with shaking, followed by sequentially decreasing the methanol concentration in the xylene and methanol mixture to 90%, 80%, and 50% for 1 hr each. After deparaffinization, specimens were stained with 80 μg/mL PI solution (Sigma) for 5 min, and then OT-TP-LSM imaging was conducted on the sectioned surface.

## Image acquisition and post-processing

Raw image datasets from dual sCMOS cameras were acquired and processed on a workstation with 128 Gb RAM and a 2 TB SSD drive. The imaging time and data size per 1 cm$^2$ area with 400 fps was 7 min and 318 GB ( = (7 × 60) s × 400 fps × (1850 × 512 × 2) byte) for each channel, respectively. The raw image strip was sheared at 45° with respect to the sample surface, and a custom image processing algorithm was used to transform the image data in the XYZ coordinate. The processing for en-face image was conducted in MATLAB and took ~1.7 s Gb$^{-1}$ after loading the image dataset at ~6.8 s Gb$^{-1}$ in the current laboratory setting. Mosaic images were generated by joining the image strips manually.

## Virtual H&E staining of OT-TP-LSM via deep learning network

CycleGAN is an unsupervised deep learning network without the need for well-aligned image pairs (*Zhu et al., 2017*) and was used for style transfer of OT-TP-LSM images to virtual H&E images (*Chen et al., 2021*; *Cao et al., 2023*; *Abraham et al., 2020*). CycleGAN model consisted of two generators (G: real OT-TP-LSM → fake H&E and F: real H&E → fake OT-TP-LSM), and two corresponding discriminators (D$_{OT-TP-LSM}$ and D$_{H\&E}$) to identify each fake image (*Figure 6—figure supplement 1*). Cycle-consistent loss was combined with adversarial loss to prevent the well-known problem of mode collapse, where all input images mapped to the same output images (*Zhu et al., 2017*). For CycleGAN training, PI-labeled OT-TP-LSM and real H&E images were collected from pancreas specimens, including each of normal, premalignant lesions, and cancer. Collected images were cropped into 512×512 pixels patch images, and augmented by applying rotation (90°, 180°, 270°) and flip (vertical, horizontal). The total training dataset was 14,322 and 22,644 patches for OT-TP-LSM and H&E, respectively. Adam solver was used as an optimizer (*Kingma and Ba, 2014*). Hyper-parameters were set with a batch size of 1 and an initial learning rate of 0.0002 and 100 epochs. Once the training was completed, generator G was used to transform OT-TP-LSM patches into virtual H&E ones. The CycleGAN training and testing were performed using a Nvidia GeForce RTX 3090 with 24 GB RAM. The network was implemented using Python version 3.8.0 on a desktop computer with a Core i7-12700K CPU@3.61 GHz and 64 GB RAM, running Anaconda (version 22.9.0). The inference time for converting OT-TP-LSM patch image into virtual H&E patch image was measured as 160ms.

## Acknowledgements

This work was supported in part by the National Research Foundation (NRF) grant funded by the Korea government (NRF-2020R1A2C3009309); and the Korea Medical Device Development Fund (KMDF) grant funded by the Korean government (Project Number: 2710002419, KMDF_RS-2023-00251073).

## Additional information

### Funding

| Funder | Grant reference number | Author |
| --- | --- | --- |
| National Research Foundation of Korea | NRF-2020R1A2C3009309 | Ki Hean Kim |

| Funder | Grant reference number | Author |
|---|---|---|
| Korea Medical Device Development Fund | 2710002419 | Ki Hean Kim |
| Korea Medical Device Development Fund | KMDF_RS-2023-00251073 | Ki Hean Kim |

The funders had no role in study design, data collection and interpretation, or the decision to submit the work for publication.

## Author contributions
Won Yeong Park, Conceptualization, Data curation, Software, Validation, Investigation, Methodology, Writing – original draft, Writing – review and editing; Jieun Yun, Software, Visualization; Jinho Shin, Byung Ho Oh, Gilsuk Yoon, Seung-Mo Hong, Resources, Formal analysis; Ki Hean Kim, Conceptualization, Supervision, Funding acquisition, Methodology, Writing – original draft, Writing – review and editing

## Author ORCIDs
Gilsuk Yoon ⓘ https://orcid.org/0000-0002-9941-024X
Ki Hean Kim ⓘ https://orcid.org/0000-0003-3410-7706

## Ethics
Fresh skin cancer specimens were collected under the Institutional Review Board (IRB) of Yonsei University, Severance Hospital, approved protocol (approval # 4-2021-0128). The study was conducted according to the Declaration of Helsinki Principles.- Formalin-fixed, paraffin-embedded (FFPE) tissue blocks of pancreatic and prostate cancers were collected under approval (approval #, 2021-0393) from the IRB of Asan Medical Center, University of Ulsan College of Medicine.

Reviewer #1 (Public Review): https://doi.org/10.7554/eLife.92614.3.sa1
Reviewer #2 (Public Review): https://doi.org/10.7554/eLife.92614.3.sa2
Author response https://doi.org/10.7554/eLife.92614.3.sa3

# Additional files

## Supplementary files
• MDAR checklist

## Data availability
All OT-TP-LSM, H&E images and CycleGAN training dataset are publicly available on Dryad (https://doi.org/10.5061/dryad.pk0p2ngwn). Code for 3D reconstruction image processing (Matlab) and virtual H&E is available on Github (https://github.com/Won-Yeong-Park/OT-TPLSM copy archived at *Park, 2024*).

The following dataset was generated:

| Author(s) | Year | Dataset title | Dataset URL | Database and Identifier |
|---|---|---|---|---|
| Park WY | 2024 | Data from: Open-top Bessel beam two-photon light sheet microscopy for three-dimensional pathology | https://doi.org/10.5061/dryad.pk0p2ngwn | Dryad Digital Repository, 10.5061/dryad.pk0p2ngwn |

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
