## [Editor Report · eLife assessment]

This **important** work by Park et al. demonstrates an open-top two-photon light sheet microscopy (OT-TP-LSM) for lesser invasive evaluation of intraoperative 3D pathology. The authors provide **convincing** evidence for the effectiveness of this technique investigating various human cancer cells. This article will be of broad interest to biologists and, specifically, pathologists utilizing 3D optical microscopy.

---

## [Referee Report · Reviewer #1 (Public Review)]

Summary:

This manuscript presents the development of a new microscope method termed "open-top two-photon light sheet microscopy (OT-TP-LSM)". While the key aspects of the new approach (open top LSM and Two-photon microscopy) have been demonstrated separately, this is the first system of integrating the two. The integration provides better imaging depth than a single-photon excitation OT-LSM.

Strengths:

- Use of liquid prism to minimize the aberration induced by index mismatching is interesting and potentially helpful to other researchers in the field.

- Use of propidium iodide (PI) provided a deeper imaging depth.

Weaknesses:

-None noted.

---

## [Referee Report · Reviewer #2 (Public Review)]

In this manuscript, the authors developed an open-top two-photon light sheet microscopy (OT-TP-LSM) that enables high-throughput and high-depth investigation of 3D cell structures. The data presented here shows that OT-T-LSM could be a complementary technique to traditional imaging workflows of human cancer cells.

High-speed and high-depth imaging of human cells in an open-top configuration is the main strength of the presented study. An extended depth of field of 180 µm in 0.9 µm thickness was achieved together with an acquisition of 0.24 mm2/s. This was confirmed by 3D visualization of human cancer cells in the skin, pancreas, and prostate.

---

## [Author Response]

The following is the authors’ response to the original reviews.

**eLife assessment**
This important work by Park et al. introduces an open-top two-photon light sheet microscopy (OT-TP-LSM) for lesser invasive evaluation of intraoperative 3D pathology. The authors provide convincing evidence for the effectiveness of this technique in investigating various human cancer cells. The paper needs some minor corrections and has the potential to be of broad interest to biologists and, specifically, pathologists utilizing 3D optical microscopy.

We would like to thank the editor for the positive general comment. We revised the manuscript by addressing the reviewers' comments.

**Public Reviews:**

**Reviewer1**
Summary:A2. This manuscript presents the development of a new microscope method termed "open-top two-photon light sheet microscopy (OT-TP-LSM)". While the key aspects of the new approach (open-top LSM and Two-photon microscopy) have been demonstrated separately, this is the first system of integrating the two. The integration provides better imaging depth than a single-photon excitation OT-LSM.Strengths:The use of liquid prism to minimize the aberration induced by index mismatching is interesting and potentially helpful to other researchers in the field.The use of propidium iodide (PI) provided a deeper imaging depth.Weaknesses:Details are lacking on imaging time, data size, the processing time to generate large-area en face images, and inference time to generate pseudo H&E images. This makes it difficult to assess how applicable the new microscope approach might be in various pathology applications.

B2. We would like to thank the reviewer for the critical and positive comments. We agree with the reviewer that detailed information such as processing time is missing.

The imaging time and data size were estimated per 1cm2 area and they were 7 min and 318 GB ( = (7 × 60) s × 400 fps × (1850 × 512 × 2) byte) for each channel, respectively. The time for processing en-face images was relatively long by taking ~1.7 s Gb−1 after loading the image dataset at ~6.8 s Gb−1 in the current setting and needs to be shortened for intraoperative application. The time for converting OT-TP-LSM images of 512 x 512 pixels into virtual H&E staining images was 160 ms. This study was to address the current limitation of 3D pathology such as imaging depth and to develop the image processing to generate virtual H&E images. Further development such as speeding up the image processing would be needed. We added missing information and included some discussion on limitations of the new system and further development for intraoperative applications.

C1-1. Revised manuscript, Discussion, pages 14-15 and lines 320-328

Although OT-TP-LSM enabled high-speed 3D imaging, the post-processing time of the OT-TP-LSM image datasets was relatively long due to the large data size, sequential processing of dual channel images, and manual stitching. The long post-processing time needs to be resolved for intraoperative applications. To speed up processing, these processing steps can be performed using field-programmable gate array (FPGA)-based data acquisition with graphics processing unit (GPU)-based computing. The processing time can be further reduced by coding the algorithm in a C++-based environment. Furthermore, ImageJ-based software such as the Bigstitcher plugin can be used for automatic 3D image processing [44].

C1-2. Revised manuscript, Materials and methods, Image acquisition and post-processing, page 17 and lines 390-398

Image acquisition and post-processing

Raw image datasets from dual sCMOS cameras were acquired and processed on a workstation with 128 Gb RAM and a 2 TB SSD drive. The imaging time and data size per 1cm2 area with 400 fps was 7 min and 318 GB ( = (7 × 60) s × 400 fps × (1850 × 512 × 2) byte) for each channel, respectively. The raw image strip was sheared at 45° with respect to the sample surface, and a custom image processing algorithm was used to transform the image data in the XYZ coordinate. The processing for en-face image was conducted in MATLAB and took ~1.7 s Gb−1 after loading the image dataset at ~6.8 s Gb−1 in the current laboratory setting. Mosaic images were generated by joining the image strips manually.

C1-3. Revised manuscript, Materials and methods, Virtual H&E staining of OT-TP-LSM via deep learning network, page 18 and lines 414-418

The CycleGAN training and testing were performed using a Nvidia GeForce RTX 3090 with 24 GB RAM. The network was implemented using Python version 3.8.0 on a desktop computer with a Core i7-12700K CPU@3.61 GHz and 64 GB RAM, running Anaconda (version 22.9.0). The inference time for converting OT-TP-LSM patch image into virtual H&E patch image was measured as 160 ms.

**Reviewer 2**
Summary:A2. In this manuscript, the authors developed an open-top two-photon light sheet microscopy (OT-TP-LSM) that enables high-throughput and high-depth investigation of 3D cell structures. The data presented here shows that OT-T-LSM could be a complementary technique to traditional imaging workflows of human cancer cells.Strengths:High-speed and high-depth imaging of human cells in an open-top configuration is the main strength of the presented study. An extended depth of field of 180 µm in 0.9 µm thickness was achieved together with an acquisition of 0.24 mm2/s. This was confirmed by 3D visualization of human cancer cells in the skin, pancreas, and prostate.Weaknesses:The complementary aspect of the presented technique in human pathological samples is not convincingly presented. The traditional hematoxylin and eosin (H&E) staining is a well-established and widely used technique to detect human cancer cells. What would be the benefit of 3D cell visualization in an OT-TP-LSM microscope for cancer detection in addition to H&E staining?

B2. We would like to thank the reviewer for the critical and positive comments. 3D pathology has been a long-standing research direction. The current pathology is 2D by examining H&E histology slides which were generated by thin sectioning biopsied and surgical specimens at different depths. The reliability of the pathological diagnosis suffers from under sampling of specimens. Although 3D pathology is possible by serial thin-sectioning, imaging, and then combining the images in 3D, it is not practice for clinical use due to the required labor and time.

We demonstrated the advantages of OT-TP-LSM in various human cancer tissues. The relatively high imaging depths of OT-TP-LSM enabled the nondestructive visualization of detailed 3D cell structures with high contrast and without distortion and allowed a distinction between cancer and normal cell structures as well as the detection of cancer invasiveness within tissues. We revised the manuscript to explain the benefits of 3D pathology with OT-TP-LSM.

C2-1. Revised manuscript, Results, 3D OT-TP-LSM imaging of human skin cancers, pages 8-9 and lines 176-180

Using 3D visualization, normal glandular structures in the dermis were distinguished from BCC tumor nests (Video 1). Both eccrine and sebaceous glands could appear similar to BCC nests in 2D images at certain depths. Hence, nondestructive 3D visualization of cell structures would be important for distinguishing them, serving as a complement to the traditional 2D H&E images.

C2-2. Revised manuscript, Results, 3D OT-TP-LSM imaging of human pancreatic cancers, pages 10-11 and lines 222-232

Magnified images of ROI 1 (PDAC) at two different depths showed irregularly shaped glands with sharp angles and 3D structural complexity including unstable bridging structure inside (Figure 4B). An irregular and distorted architecture amidst desmoplastic stroma is one of the important diagnostic factors for PDAC [35]. The cancer glands exhibited disorganized cancer cell arrangement with nuclear membrane distortion. Magnified images of ROI 2 showed both nonneoplastic ducts and cancer glands in different cell arrangements (Figure 4C). The nonneoplastic ducts showed single-layered epithelium with small, evenly distributed cells expressing relatively high nuclear fluorescence. Cancer glands, on the other hand, had disorganized and multilayered structure with large nuclei. OT-TP-LSM visualized the 3D invasiveness of cancer glands within tissues nondestructively, which could not be identified from limited 2D information.

C2-3. Revised manuscript, Results, 3D OT-TP-LSM imaging of human prostatic cancers, page 11 and lines 251-252

OT-TP-LSM provided histological 3D information equivalent to that of the H&E stained image without the need for sectioning.

C2-4. Revised manuscript, Discussion, page 12 and lines 274-276

OT-TP-LSM was developed for the rapid and precise nondestructive 3D pathological examination of excised tissue specimens during both biopsy and surgery, as a compliment to traditional 2D H&E pathology by visualizing 3D cell structures.

C2-5. Revised manuscript, Discussion, page 13 and lines 284-288

The relatively high imaging depths of OT-TP-LSM enabled the nondestructive visualization of detailed 3D cell structures with high contrast and without distortion and allowed a distinction between cancer and normal cell structures as well as the detection of cancer invasiveness within tissues. These have been challenging with 2D histological images.

**Reviewer #2 (Recommendations For The Authors):**
I would suggest the following points to the authors to enhance the readability of the manuscript and to provide a strong narrative to explain their findings:A3. Line 54: For the non-expert readers, please provide more background information about the histopathology before introducing the hematoxylin and eosin staining.

B3. We would like to thank the reviewer for the comment. As suggested by the reviewer, we added information about the current standard method of histopathological examination and its limitations.

C3. Revised manuscript, introduction, page 4 and lines 56-64Precise intraoperative cancer diagnosis is crucial for achieving optimal patient outcomes by enabling complete tumor removal. The standard method is the microscopic cellular examination of surgically excised specimens following various processing steps, including thin sectioning and hematoxylin and eosin (H&E) cell staining. However, this examination method is laborious and time-consuming. Furthermore, it has inherent artifacts that disturb accurate diagnosis, including tissue loss, limited two-dimensional (2D) information, and sampling error [1]. High-speed three-dimensional (3D) optical microscopy, which can visualize cellular structures without thin sectioning, holds promise for nondestructive 3D pathological examination as a complement of 2D pathology limitation [1-4].

A4. Line 66 and 71: Please briefly introduce the cited studies to give some information about the previous studies. This will help to reader to understand the innovative aspects of your study.

B4. We would like to thank the reviewer for the comment. As suggested by the reviewer, we added a brief introduction about the cited studies.

C4. Revised manuscript, introduction, pages 4-5 and lines 71-82

As a deep tissue imaging method, two-photon microscopy (TPM) has been used in both biological and optical biopsy studies [17-19]. TPM is based on nonlinear two-photon excitation of fluorophores and achieves high imaging depths down to a few hundred micrometers by using long excitation wavelengths, which reduce light scattering. Moreover, TPM provides additional intrinsic second harmonic generation (SHG) contrast for visualizing collagen fibers within the extracellular matrix (ECM). This feature proved advantageous for high-contrast imaging of cancer tissue and microenvironmental analysis [20-22]. However, TPM has low imaging speeds due to point scanning-based imaging. To address this limitation, two-photon LSM (TP-LSM) techniques were developed for high-speed imaging [23-27]. Although TP-LSM facilitated rapid 3D imaging of cancer cells and zebrafish, its applications were limited to small samples and biological studies due to geometric limitations.

A5. Line 72: Please mention the importance and benefit of having an open-top configuration. I think this is one of the key aspects that provide a high imaging depth in OT-LP-LSM.

B5. We would like to thank the reviewer for the comment. Conventional LSM techniques including TP-LSM have a configuration in which the illumination objective is oriented in the horizontal plane and imaging is performed with orthogonally arranged objectives. However, this geometry limited lateral sample size physically and it is unsuitable to image centimeter-scale large tissue. Therefore, we developed OT-TP-LSM for 3D large tissue examination. High imaging depths were achieved with long excitation wavelengths and long emission wavelengths of fluorophores. The open-top configuration does not contribute to the improvement of imaging depth. We revised the manuscript to explain the need for open-top configuration.

C5. Revised manuscript, introduction, page 5 and lines 82-86

Conventional TP-LSM had a configuration of a horizontally oriented illumination objective and a vertically oriented imaging objective. This geometry imposed limitations on the sample size, rendering it unsuitable for the examination of centimeter-scale specimens. TP-LSM with open-top configuration is needed for 3D histological examination.

A6. Line 78: It would be nice to clearly quantify the imaging depth here.

B6. We would like to thank the reviewer for the comment. Although we considered entering the quantitative imaging depth of OT-TP-LSM in the introduction section, we decided that it would be appropriate to present the quantitative imaging depth in the Results section and discuss it in the Discussion section.

A7. Line 146: Please clearly explain the reason why the upper layers are not resolved.

B7. We would like to thank the reviewer for the comment and we are sorry for the missing information. The skin epidermis has various cell layers and superficial layers are composed of less rounded and flat cells with relatively small cytoplasm. Therefore, cells in that layer could be difficult to resolve with the current system resolution because there is little space between nuclei. Additionally, strong autofluorescence signal in the stratum corneum could be the reason for preventing visualization of the cells in the superficial layer. We revised the manuscript to explain the reasons in detail.

C7. Revised manuscript, Results, 3D OT-TP-LSM imaging of human skin cancers, page 8 and lines 159-163

Keratinocytes in the basal layer were relatively large and individually resolved, while those in the upper layers were unresolved and appeared as a band. It could be attributed to the upper layers being comprised of flat cells with relatively small cytoplasm, resulting in little space between nuclei. Additionally, strong autofluorescence signal in the stratum corneum might prevent visualization of the cells in the superficial layer.

A8. Line 253: Please explain the importance of visualization of 3D cell structures in cancer pathology. I think this should be stated clearly throughout the text as it is the key component of OT-LP-LSM to complement the traditional H&E staining. Also, referring to the non-destructive manner of your technique would help to emphasize this point.

B8. We would like to thank the reviewer for the comment. As answered in A2, the current H&E histological examination has inherent limitations due to limited 2D information and sampling errors. To resolve this, OT-TP-LSM was developed for the visualization of 3D cell structures nondestructively as a complement to traditional slide-based 2D pathology. We demonstrated the advantages of OT-TP-LSM in various human cancer tissues. The relatively high imaging depths of OT-TP-LSM enabled the nondestructive visualization of detailed 3D cell structures with high contrast and without distortion and allowed a distinction between cancer and normal cell structures as well as the detection of cancer invasiveness within tissues. We revised the manuscript to explain the benefits of 3D pathology with OT-TP-LSM.

C8. Please refer to the answer in C2-1 – C2-5.

A9. Figures: Please clearly mark the cancer regions in the images as indicated in Figure 5. It will help the reader to easily compare the healthy and invaded tissue parts.

B9. We would like to thank the reviewer for the comment. We confirmed that the cancer area is not marked in Figure 4 of the pancreatic cancer tissue. We modified Figure 4 to mark the cancer region. Additionally, Figure 2 of the skin cancer tissue was also modified in this regard.

C9. Modified Figure 2 and Figure 4.

**Author response image 2. sa3fig2:**